# Collaborative Filtering to Predict Sensor Array Values in Large IoT Networks

**DOI:** 10.3390/s20164628

**Published:** 2020-08-17

**Authors:** Fernando Ortega, Ángel González-Prieto, Jesús Bobadilla, Abraham Gutiérrez

**Affiliations:** Departamento de Sistemas Informáticos, Escuela Técnica Superior de Ingeniería de Sistemas Informáticos, Universidad Politécnica de Madrid, 28031 Madrid, Spain; fernando.ortega@upm.es (F.O.); angel.gonzalez.prieto@upm.es (Á.G.-P.); jesus.bobadilla@upm.es (J.B.)

**Keywords:** collaborative filtering, sensor arrays, IoT, matrix factorization

## Abstract

Internet of Things (IoT) projects are increasing in size over time, and some of them are growing to reach the whole world. Sensor arrays are deployed world-wide and their data is sent to the cloud, making use of the Internet. These huge networks can be used to improve the quality of life of the humanity by continuously monitoring many useful indicators, like the health of the users, the air quality or the population movements. Nevertheless, in this scalable context, a percentage of the sensor data readings can fail due to several reasons like sensor reliabilities, network quality of service or extreme weather conditions, among others. Moreover, sensors are not homogeneously replaced and readings from some areas can be more precise than others. In order to address this problem, in this paper we propose to use collaborative filtering techniques to predict missing readings, by making use of the whole set of collected data from the IoT network. State of the art recommender systems methods have been chosen to accomplish this task, and two real sensor array datasets and a synthetic dataset have been used to test this idea. Experiments have been carried out varying the percentage of failed sensors. Results show a good level of prediction accuracy which, as expected, decreases as the failure rate increases. Results also point out a failure rate threshold below which is better to make use of memory-based approaches, and above which is better to choose model-based methods.

## 1. Introduction

Sensor arrays are groups of sensors working in the same aim, and usually deployed in some spatial pattern, including circular, planar, linear, spherical and cylindrical shapes. The sensor arrays main advantage is the new dimensions that they provide to the observation. Usual applications of array sensors are signal-to-noise ratio gain, estimation of the direction of the signal and parameter prediction. This later parameter estimation is a key application of the array sensors, since it takes advantage of the spatial or temporal properties of the incoming signals. Research in array sensors is currently on the rise and it covers a wide range of areas, such as medical, chemical, communication and Internet of Things (IoT).

Current research in medical applications of sensor arrays makes use of several base sensors. For instance, an e-nose system has been developed based on metal oxide sensor arrays [1] to distinguish diabetic patients from healthy individuals. The e-nose system incorporates the sensor array and the data acquisition stage, and processing includes clustering and dimensionality reduction by means of the Principal Component Analysis (PCA) method. A related research is presented in [2], where a sensor array is used to discriminate diabetes: acetone presence has been reported in the diabetic patient breath, and results show that control groups and patients can be differentiated by means of it. Another e-nose array of sensors experience has been designed to detect ammonia concentration in breath by means of a conductometric array of eleven different polyaniline nanocomposite sensors [3]; authors have chosen the sensor array approach due to the drift and sensitivity to humidity sensor weakness. An array of colorimetric sensors has been used for detecting extremely small amounts of target molecules [4], so that it analyzes a pattern of color change caused by a reaction between the sensor array and external substances. The biosensor array could detect hormone drugs (estrogen) and antibiotics.

Sensor arrays have also been applied to other domains. In [5], a driver safety experience has been based on the use of a pressure sensor array placed in the backrest of the seat, for which results show a high correlation between heartbeat and electrocardiogram signals. They have also been used in [6] for improving wind turbines performance, which depends on wind speed and direction. To measure these wind parameters in 3D, the authors used a semi-conical ultrasonic sensor array, and simulation results validate the accuracy and anti-noise performance of the proposed method. In addition, a low-cost sensor array (AIRQino) has been used [7] to monitoring atmospheric composition and meteorological samples collected in the Artic (${\mathrm{CO}}_{2}$, air temperature and humidity, particulate matter), and the experience validates the use of the sensor array in this extreme environment. There exists also proposals that make use of deep learning through recurrent neural networks, for improving the accuracy in the concentration identification of gas mixtures in gas sensor array raw data, as in [8]. Finally, feature selection can be a main stage in the sensor arrays data analysis: in [9] the authors make use of an e-nose applied to odor-detection and they report that this task requires adequate extraction and selection of features that will be used in the final model.

Arrays of sensors approaches are also on the rise in the chemical industry field, particularly to detect a variety of gases. Three toxic gases (CO, ${\mathrm{NO}}_{2}$ and ${\mathrm{NH}}_{3}$) have been differentiated by means of a novel Quartz Crystal Microbalance (QCM) sensor array [10] and PCA has been used to extract the target toxic gas under testing conditions. In addition, in [11], from an array of eight chemoresistive gas sensors, dimensionality reduction techniques have been used to classify, using the K-Nearest Neighbors algorithm, from $H_{2}$, ${\mathrm{CH}}_{4}$ and CO gases. In [12], some volatile compounds in food samples are detected by means of an array of sensors using polypyrrole-zinc oxide, by using the Central Composite Design (CCD). A sensitive, low-cost, sensor array method is presented in [13]: it performs the analysis of metal ions and oxyanion based on the ultraviolet and visible spectra of amino acid-gold nanoparticles. Furthermore, photovoltaic arrays play a fundamental role in solar plants, environments in which the correct measure of temperature is critical. In this vein, [14] quantifies the uncertainty of the operating temperature of large photovoltaic arrays as a function of the location, type and number of the sensors.

Nevertheless, one of the most important present and future applications of sensor arrays systems are towards the Internet of Things (IoT). For instance, traffic congestion control is conducted in [15] by means of sensors group to improve the traffic efficiency. IoT medicine applications are also growing in the sensor arrays research field; an exhaustive heart activity monitoring is performed in [16] by using arrays of inductive sensors. Chemical IoT is made through sensor arrays, such as in [17], where nanowire metal oxide semiconductors have been developed to detect anomalous industrial discharges and their spatial location.

Despite of their enormous success, IoT approaches are forced to deal with some intrinsic issues derived of their omnipresence. IoT is usually based on very large amounts of sensors deployed as sensor arrays, potentially covering very large geographically areas. In this way, some sensors may be located in extreme weather places whereas some others always work in controlled conditions, each sensor array readings are sent to the cloud using networks with different Quality of Service (QoS), and there may exist sensors with outdated versions of the firmware. For these reasons, it is unavoidable to be forced to deal with a non-negligible failure rate in the sensors network, in such a way that the matrix of sensor readings will present a certain sparsity level. In this way, it is crucial for IoT research to be able to manage this failures and to provide accurate predictions of the missing readings. To mention some of the existing proposals in the literature, in [18] the authors tackle imperfect transmission between the system and the controller through a mode-dependent stochastic measurement fading model. The fault-tolerant robust non-fragile H∞ filtering problem for networked control systems with sensor failures is studied in [19]. Finally, vehicle motion sensors failure is detected and recovered in [20], where the authors show that the fault detection and restoration algorithm can satisfy the control performance for each sensor failure. In addition, there exists proposals in the literature based on using classical machine learning approaches to process sensor arrays values, like in [8,10,18].

Source reconstruction from sensor arrays have been reported in several publications. Source localization has been made from sparse representation of sensor measurements [21] by means of Singular Value Decomposition. A Bayesian algorithm for Detection of Arrival (DoA) has been proposed in [22] that can calibrate the unknown errors in the sensor array and estimate the DOA of the sources simultaneously. By exploiting the joint spatial and spectral correlations of acoustic sensor array data, ref. [23] provides DoA estimations using sampled acoustic data. They also investigate the spatial sparsity in a linear sensor array configuration. In addition, very recently a deep transfer learning architecture has been used for sparse sensor array selection by means of a Convolutional Neural Network (CNN) [24]. Finally, to get a balance between resolution and hardware cost, in [25] sparse subarrays are chosen based on the target scenario. To be precise, authors design a CNN that acts on the array covariance matrix and selects the best sparse subarray that minimizes the error.

However, these works focused on dealing with sensor failures are intrinsically based on methods that expect non-sparse inputs, so these are definitely not appropriate approaches to process sparse matrices of sensor values, as the ones expected from sensors failures. In order to address this problem, we propose to apply techniques imported from the Recommender Systems (RS) [26] field to face the sparsity issue; more specifically based on its Collaborative Filtering (CF) [27] facet. CF RS are designed to provide item recommendations (say of movies, songs or products in general) to users based on their preferences, by means of the ratings the user scored to the items he/she consumed. However, an average user only consumes, purchases, and thus votes, a very reduced number of the existing items. Hence, the matrices that gather these ratings are typically very sparse as they contain a limited number of rating values from the whole set of possible ratings, of dimension number of users times number of screened items. For this reason, current CF RS machine learning methods are designed to provide accurate predictions and recommendations based on sparse matrices of ratings [28], so they provide a bunch of techniques that, presumably, will be well-suited for dealing with sensor failures.

The key idea of our proposal is that, fortunately, sensor array results are not completely independent from each other; e.g., readings from sensors under high environmental temperature in some area in China can help to predict readings of some other sensors that have reported failures in a high temperature area from Nevada; it can be done when a correlation in the sensor values of both areas occurs. This is precisely what CF RS are trained for: they predict item’s values that the users have not voted yet based on the rating values of the rest of the RS users. The analogy with the IoT scenario occurs when sensors play the role of RS items, and each instant of time in which the sensor array was measure plays the role of an RS user. In this case, we can predict some missing sensor value from the whole set of measures of the IoT network along time and space, in the same way that we predict some item value from the whole set of ratings in the RS. Figure 1 shows the explained concept: when sensor arrays are deployed in large IoT networks, each failed sensor value can be predicted from the whole set of sensor array values in the system.

In this paper, we will consider some of the most widely used CF RS methods. Historically, the first one that was applied to RS was the celebrated K-Nearest Neighbors (KNN) method [26], and it has been used as reference and baseline to this research. The idea is very simple: to predict the missing values by mimicking the ratings of the most similar users or items. For this reason, KNN approaches are typically classified as user-based and item-based ones [29].

Nevertheless, whereas KNN is a memory-based method, currently model-based methods dominate the scene. In this family, the most used CF methods are the Matrix Factorization (MF) based ones [30], although recently the deep learning approaches start gaining importance in the field [31,32]. The idea of the MF method is to make a dimensional reduction of the RS sparse rating matrix, decomposing it into two lower-dimensional dense matrices. In this way, each user and each item are coded using a low number of hidden factors, all of them lying in the same vector space, in such a way the predictions can be obtained by inner product of the hidden user factors against the hidden item factors. It is worthy to mention that, compared with the white-box KNN, this is a black-box method, although there exist novel deep learning approaches to unhide factors [33]. Moreover, there are several MF variations used in CF. Probabilistic Matrix Factorization (PMF) [34] and Biased MF [35] are the base factorization algorithms. The non-Negative MF (NMF) [36], where ratings cannot be negative, is an MF method that helps to understand the hidden semantic of the factors. In the Bayesian non-Negative Matrix Factorization (BNMF) [37], factors lie within the range [0, 1]. It operates by grouping the users that share the same tastes and it provides an understandable probabilistic meaning to the hidden factors. In this way, in this paper we will use some KNN and MF methods as baselines to test the proposed approach.

The hypothesis of the paper is that CF techniques can be used to accurately predict failed sensor values from the whole set of sensor readings collected from the IoT network. The concept is borrowed from the RS research field, where rating matrices are really sparse. Since sparsity levels are much smaller in the IoT scenario than on the RS field, the testing CF methods will cover both memory-based methods (KNN) and model-based methods (MF). According to the RS research experiences, the expected results point towards (a) an accurate level of prediction accuracy, (b) a decrease of accuracy as the sensors failure ratio increases, and (c) a better performance of the model-based approaches when the sensors failure ratio is not low.

The rest of the paper has been structured as follows: in Section 2 the proposed method is explained, and the experiments design is defined. Section 3 shows the experimental results and their discussions. Finally, Section 4 contains the main conclusions of the paper and the future works.

## 2. Materials and Methods

The aim of the proposed approach is to make use of CF RS methods and algorithms in order to accurately predict failed values from large IoT networks of sensor arrays. Figure 2 shows the details: the CF RS field (left side in Figure 2) make recommendations (highest predictions) to users, based on the set of voted, consumed, listened to, etc. items. Data can conceptually be stored in large sparse matrices of ratings, such as the MovieLens, Netflix or Last.fm databases, where users are arranged in rows and items are arranged in columns. Figure 2 shows this matrix, where numbers represent votes and empty cells represent non voted items. The same concept can be applied to the IoT networks, where different types of sensors play the role of items and the existing set of sensor measures along time play the role of users. Figure 2 shows the IoT matrix where the sparsity is not as high as the RS one, since sensor failure situations will be much less frequent than not consumed items in an RS. Please note that this difference can produce a significant impact on the performance of the existing CF methods: they are designed to operate on matrices with a representative level of sparsity. A main objective of this paper is precisely to test the quality of state-of-the-art CF methods when applied to the less sparse IoT context.

The right side of Figure 2 explains the MF operation. It creates two dense matrices of a reduced common dimension (the hidden factors): the number of factors is less than the number of sensors and much less than the number of sensor measures in the IoT network. Once the iterative MF algorithm has learned the factors, we can predict the value of a failed sensor j in an instant of time i by making the dot product of their factors. It is important to realize that hidden factors capture the essence of the sparse matrix values, and consequently predictions are made based on the whole set of reading values in the IoT sparse matrix. As an example: the value of a sensor failed due to excessive UV radiation can be predicted from the values of a set of similar not failed sensors under similar extreme situations at other moment of time.

### 2.1. Collaborative Filtering Methods

Since the sparsity of the IoT failures is not as high as in CF datasets, we have selected both memory-based and model-based methods as baselines to make the experiments. From the memory-based algorithms we have chosen the representative KNN method based on items [26] and the KNN based on users [29]. From the model-based methods we have chosen the PMF [34], BiasedMF [35] and NMF [36] matrix factorization ones. Both KNN algorithms have been chosen because memory-based methods perform better in non highly sparse scenarios, such as the IoT failures one [28]. PMF and BiasedMF have been selected because they are the baseline model-based collaborative references [30]. Finally, NMF is particularly relevant when the sensor arrays provide non negative values; in these cases NMF hidden factors can be used to extract latent semantic information [38].

PMF is an archetypal MF method. For the convenience of the reader, we sketch briefly its formulation. For further information, check the aforementioned references. As in any CF based RS, as initial data we have a sparse rating matrix $R=(r_{ui})$ that contains the known ratings that the user *u* emitted about the item *i*, denoted $r_{ui}$. This is a matrix of dimension number of users times number of items, and that only contains very few known ratings. The aim of the MF method is to fill the gaps of *R* as best as possible to obtain the unknown ratings, from which a recommendation can be issued.

The key idea of the method to accomplish this is to factorize *R* into two lower dimension matrices *P* and *Q*, being *P* the matrix of users and *Q* the matrix of items, in such a way that
$$R\approx \hat{R}=P\cdot Q^{t}.$$

Both matrices have a common dimension *k* (the number of hidden factors), where *k* is typically much smaller than the number of users and items. When the model has learnt, each user *u* is represented by means of a *k*-dimensional vector $p_{u}=\left(p_{u,1},\ldots ,p_{u_{k}}\right)$, and each item *i* is represented by a *k*-dimensional vector $q_{i}=\left(q_{i,1},\ldots ,q_{i,k}\right)$, that are stored as the *u*-th row of *P* and the *i*-th column of *Q*, respectively. In this way, the prediction ${\hat{r}}_{ui}$ of the item *i* to the user *u* is obtained by making the dot product
$${\hat{r}}_{ui}=p_{u}\cdot q_{i}=\sum_{l=1}^{k}p_{u,l}q_{i,l}.$$

The users’ and the items’ hidden factors belong to the same vector space, and they share the same semantic: when their factors line up, predictions are relevant.

In order to find the appropriate latent factors $p_{u}$ and $q_{i}$, we seek to minimize the regularized squared cost function
$$f\left(p_{u},q_{i}\right)=\sum_{r_{ui}\neq •}{\left(r_{ui}-{\hat{r}}_{ui}\right)}^{2}+\lambda \left(\left|\right|p_{u}{\left|\right|}^{2}+\left|\right|q_{i}{\left|\right|}^{2}\right)=\sum_{r_{ui}\neq •}{\left(r_{ui}-p_{u}\cdot q_{i}\right)}^{2}+\lambda \left(\left|\right|p_{u}{\left|\right|}^{2}+\left|\right|q_{i}{\left|\right|}^{2}\right)$$

Here, $r_{ui}\neq •$ means that the rating $r_{ui}$ is known, that is, that the user *u* did vote the item *i*; and λ>0 is a constant that tunes the strengh of the regularization term.

By applying a standard optimization procedure by gradient descent to *f*, it is straightforward to obtain an iterative minimization method. To be precise, if $p_{u}^{t}$ and $q_{i}^{t}$ denote the corresponding approximations of the optimal values of $p_{u}$ and $q_{i}$ at the *t*-th step, the associated update rule is, for any pair (u,i) with $r_{ui}\neq •$
$$\begin{array}{cc} p_{u}^{t+1} & =p_{u}^{t}+\gamma \left(e_{ui}^{t}q_{i}^{t}-\lambda p_{u}^{t}\right),\\ q_{i}^{t+1} & =q_{i}^{t}+\gamma \left(e_{ui}^{t}p_{u}^{t}-\lambda q_{i}^{t}\right), \end{array}$$
where γ is the gradient descent step hyper-parameter (i.e., the learning rate) and $e_{ui}^{t}=r_{ui}-p_{u}^{t}\cdot q_{i}^{t}$ is the error attained at time *t*.

To test the quality of the results, we have chosen the Root Mean Square Error (RMSE) quality measure; it measures the squared differences between target values and predicted ones. It is given by
$$\mathrm{RMSE}=\sqrt{\frac{1}{\#R^{\mathrm{test}}}\sum_{\left(u,i\right)\in R^{\mathrm{test}}}{\left(r_{ui}-p_{u}\cdot q_{i}\right)}^{2}},$$
where $R^{\mathrm{test}}$ is the collection of user-items pairs (u,i) on which we compute the quality measure (the test split of the dataset), and $\#R^{\mathrm{test}}$ denotes its cardinal.

### 2.2. Datasets Description

To test this paper’s hypothesis, we have used two open sensor array datasets collected from Kaggle [39,40], and also a synthetic dataset. With respect to the real datasets, the first one [39] stores the acquired values from 16 chemical sensors exposed to gas mixtures at varying concentration levels: Ethylene and Methane in air, and Ethylene and CO in air. The second real dataset [40] makes use of the same sensor array, generating a feature vector containing 8 individual features extracted from each particular sensor, giving rise to 128 sensor measures at a time. The resulting features from the sensors have been normalized with a min-max normalization with values in [0, 1] to feed the machine learning algorithms involved in the CF task.

In order to glace over the data, the measures have been plotted to analyze their distributions. As an example, Figure 3 shows the result for sensors 1 to 8 of the first dataset [39] (before normalization). A relevant issue is the sensors’ range of values, since it is convenient that a subset of them have similar normalized values, such as we can observe in Figure 4. For computational reasons, we have used a subset of [39], containing 84,165 random samples of time measures from the original data (1,346,640 sensor measures in total). The second dataset [40] was completely used with a total of 13,910 time measures (1,780,480 sensor measures in total).

The CF MF model discovers the complex relations existing among feature values and stores them into the hidden factors resulting from the dimensionality reduction process. For the MF to work properly it is necessary that there exists quite strong underlying relations between features. CF datasets contain a variety of correlations, since each user has a set of “semantic neighbors” that share preferences on items; analogously, each item has a set of similar items that have been voted in a similar way. However, at a first sight, it is not clear that in our sensor array datasets each sensor measures correlate with the ones of some other sensors. Since this is a key issue to fulfill the paper’s hypothesis, we have obtained the correlation matrices for both the [39] dataset (Figure 5) and the [40] dataset (Figure 6). Recall that, given two random variables *X* and *Y* (say, the time series of two sensor readings), the correlation between *X* and *Y*, $\rho_{X,Y}$, is given by
$$\rho_{X,Y}=\frac{E\left(\left(X-\mu_{X}\right)\left(Y-\mu_{Y}\right)\right)}{\sigma_{X}\sigma_{Y}}.$$

Here, $\mu_{X}$ and $\mu_{Y}$ denote the means of *X* and *Y*, $\sigma_{X}$ and $\sigma_{Y}$ their standard deviation, and E stands for mathematical expectation. In the particular case that we draw samples $x_{1},\ldots ,x_{n}$ and $y_{1},\ldots ,y_{n}$ from *X* and *Y* (say, the data from the dataset), we can estimate
$$\mu_{X}=\frac{1}{n}\sum_{i=1}^{n}x_{i},\;\mu_{Y}=\frac{1}{n}\sum_{i=1}^{n}y_{i},\;\sigma_{X}^{2}=\frac{1}{n}\sum_{i=1}^{n}{\left(x_{i}-\mu_{X}\right)}^{2},\;\sigma_{Y}^{2}=\frac{1}{n}\sum_{i=1}^{n}{\left(y_{i}-\mu_{Y}\right)}^{2},$$
and finally
$$\rho_{X,Y}=\frac{1}{\sigma_{X}\sigma_{Y}}\sum_{i=1}^{n}\left(x_{i}-\mu_{X}\right)\left(y_{i}-\mu_{Y}\right).$$

As it can be seen from Figure 5 and Figure 6, there is a high degree of correlation among the majority of our features (sensors), particularly in the [39] dataset. It is worthy to mention that the obtained correlations show a mosaic pattern, in which some groups of sensors attain high intra-group correlations, with smaller inter-group correlations. This reflects the fact that the sensors are naturally aggregated in several clusters based on the measured gas and operation characteristics.

For instance, in [39] (Figure 5), we find that sensors 3, 4, 7, 8, 11, 12, 15, 16 form a highly intra-correlated cluster, with strong positive correlations. There exists another cluster formed by the sensors 1, 5, 6, 9, 10, 13, 14 with smaller positive correlation inter and intra-cluster. On the other hand, sensor 2 presents a much weaker correlation with the other sensors (except maybe with 9 and 10) which suggests that sensor 2 has a different nature than the other ones in the dataset. Similar patterns can be observed in [40] (Figure 6), where, roughly speaking, sensors 1–16 and 65–80 seem to form a highly intra-correlated cluster, with lower inter-correlation. Observe that, due to the organization of the dataset, readings 1–16 correspond to the 8 measured features of the first 2 sensors, while readings 65–80 correspond to the 8 features of sensors 9 and 10. In addition, as in [39], we also observe isolated sensors, like sensors 10, 21 or 29, with very low absolute correlation with any other sensor.

### 2.3. Dataset Syntheses Via Copulas

Additionally, we propose to test the performance of CF RS models on a synthetic dataset. This allows us to address two intrinsic problems of IoT datasets. The first and most obvious one is that there exist very few large public datasets with rich sensor data that are suitable for testing RS. Typically, the sensor data are collected privately by the providing service companies and they do not share publicly the obtained information, even for academic research. The second one is that real sensor datasets may contain a significant proportion of outliers: it can distort the statistical information of the dataset and fake the results in such a way that very ill-posed models may attain quite good average performance due to the influence of these outliers. For this reason, testing against a synthetic dataset allows us to straighten the reliability of our conclusions: if a certain CF RS method achieves good results on both real and synthetic datasets, we have a much stronger statistical evidence that the results are sound.

The synthetic dataset has been generated as follows. Suppose that we have, as base dataset, a collection of measures of *n* sensors at different moments of time. This information can be stored as a random vector
$$X=(X_{1},X_{2},\ldots ,X_{n}),$$
where the values $X_{i}$ are real-valued random variables modeling the measure of the i-th sensor. Typically, the random variables $X_{i}$ are very far from being independent, as shown in the correlation matrices of Figure 5 and Figure 6 (indeed, this is the key point of ML methods, to exploit the correlation between the different sensors). Hence, the vector X is not fully characterized by the marginal distributions of each $X_{i}$, but by a certain unknown joint distribution. Let us denote by $F_{X_{i}}\left(x\right)=P\left(X_{i}\leq x\right)$ the marginal cumulative distribution function of the sensor $X_{i}$. Associated to the random vector X, we can consider the function given by
$$C\left(u_{1},u_{2},\ldots ,u_{n}\right)=P\left(F_{X_{1}}\left(X_{1}\right)\leq u_{1},F_{X_{2}}\left(X_{2}\right)\leq u_{2},\ldots ,F_{X_{n}}\left(X_{n}\right)\leq u_{n}\right),$$
for any real numbers $u_{1},\ldots ,u_{n}\in R$. Recall that the random variable $F_{X_{i}}\left(X_{i}\right)$ is the so-called Smirnov transformation of $X_{i}$ and it converts any random variable into a uniform distributed one. In this way, *C* is nothing but the joint cumulative distribution function of the associated uniformly-distributed Smirnov transformations of the original variables.

This *C* defines a real-valued function of several variables, $C:R^{n}\to R$, called the copula of X. It completely determines the random vector X by Sklar’s theorem [41], provided that the marginal cumulative distribution functions $F_{X_{i}}$ are continuous. Indeed, the joint cumulative distribution function, $F\left(x_{1},\ldots ,x_{n}\right)=P\left(X_{1}\leq x_{1},\ldots ,X_{n}\leq x_{n}\right)$, can be recovered in terms of the copula by
$$F\left(x_{1},\ldots ,x_{n}\right)=C\left(F_{X_{1}}\left(x_{1}\right),\ldots ,F_{X_{n}}\left(x_{n}\right)\right).$$

This allows us, once computed the copula *C*, to draw as many random samples as desired, according to the unknown underlying distribution of X. Particularly, using this idea we can synthesize a new dataset only by means of the copula *C*. For further information about the use and properties of copulas, please check [41].

In our particular case, we applied this procedure to generate a synthetic dataset from the first real dataset [39]. Recall that this baseline contains the measures of an array of 16 sensors at different times. We estimated the copula *C* from a subsample of 1000 randomly chosen instances of the dataset. From it, a fully synthetic dataset with 101,000 instances was created. All the calculations were conducted using the Python copulas public library. Observe that, by construction, the correlation matrix and marginal histograms are very similar to the ones of Figure 3 and Figure 5.

## 3. Experimental Results

As we mentioned in the previous section, three datasets are tested to evaluate the proposed approach: two real ones [39,40] and a synthetic one generated from [39]. In order to obtain a broad view of the performance of different methods, we use several baselines: KNN based on items [26] (ColKNN), KNN based on users [29] (RowKNN), Probabilistic Matrix Factorization (PMF) [34], Biased MF [35] (BiasedMF), and Non-negative Matrix Factorization [36] (NMF). The chosen quality measure is the Root Mean Square Error (RMSE), and it returns the prediction error of each algorithm on the test split.

In order to measure the sensor failure impact in the quality of predictions, we consider nine possible sparsity rates, from 0.1 to 0.9 with step 0.1. From each of the three datasets, nine sparse training versions have been obtained by removing measurements from the original dataset until reaching the desired sparsity rate. The readings to be removed have been randomly drawn through a uniform distribution. The removed values have been used for creating the corresponding nine testing datasets. On each of these derived datasets, the five proposed RS methods have been tested, giving rise to a total of 3 (datasets) × 9 (sparsitylevels) × 5 (RSmethods) = 135 conducted trials.

As it is customary in machine learning, the selected CF methods depend on a collection of hyper-parameters that impact in the learning process and must be tuned. As explained in Section 2.1, all the chosen MF-based methods (i.e., PMF, BiasedMF and NMF) depend on the number of hidden factors to be computed (*k* in the notation of Section 2.1). In addition, PMF and BiasedMF also require to optimize the learning rate, γ, and the strength of the regularization term, λ. For the KNN-like methods (i.e., RowKNN and ColKNN), the only hyper-parameter is the number of neighbors to be combined to create the prediction. In this way, for each of the aforementioned 135 experiments, we have made use of a grid search optimization to obtain the most adequate hyper-parameter values (i.e., the combination of hyper-parameters that provides the lowest RMSE). Table 1, Table 2 and Table 3 contain the experimental settings of this optimization.

Fixed the optimal hyper-parameters for each combination of dataset, baseline algorithm and sparsity level, we compare the prediction errors on the test split. Figure 7 shows the results on the real datasets: the [39] dataset is on the left-side of Figure 7, and the [40] dataset is in its right-side.

From both plots in Figure 7 we can observe that, as expected, the higher the level of sparsity the higher the error, since the CF methods have less information to make predictions. Model-based methods (PMF, BiasedMF and NMF) return similar quality values, except PMF that is more sensitive to data sparsity. The quality results of PMF collapse in the smallest dataset [39] (blue line at the left side in Figure 7) when high sparsity values (0.8 and 0.9) are tested. The memory-based RowKNN method provides the best results in the larger dataset [40] (right side of Figure 7) for small sparsity, but it does not adequately predict on high sparsity scenarios. This is consistent with the CF literature [26,28]. The ColKNN method turns out to be not competitive in any case, with high (but almost constant) error for all the sparsity levels.

Overall, the BiasedMF and the NMF model-based methods provide the best quality results in IoT scenarios where sensor failure ratings are high, whereas the RowKNN memory-based method is more adequate when the sensor failure ratings are moderate. In the tested datasets, the 0.7 sparsity threshold indicates the convenience of applying CF memory-based or model-based methods.

To confirm the claimed observations, an extra testing have been conducted on the synthetic dataset generated from [39], whose prediction errors are plotted in Figure 8. As it can be seen, the sparsity versus prediction error trends follow the same behavior than in the corresponding real dataset. Incidentally, a slightly better performance of the model-based methods is observed compared with the memory-based ones. This experiment evidences the soundness of the observed results with the real datasets. The performance of the proposed RS CF methods is independent of the particular data, and they are actually exploiting the underlying statistical information.

In this way, the observed results of the tested methods on the synthetic dataset strongly suggests that the proposed approach in this paper can be easily generalize to other sensor array sample, giving rise to highly scalable solutions for the IoT.

## 4. Conclusions

In this paper, we have addressed the problem of predicting missing values from sensor arrays deployed in large IoT systems, produced due to unreliable sensors, extreme environmental situations, deficient QoS and many other problematic issues. For that goal, we have proposed a novel approach based on the application of collaborative filtering techniques for treating the data arrays as a dataset of ratings, matching each sensor and instant of time with its corresponding reading.

It is worthy to mention that, despite it seems apparent that collaborative filtering is be useful for filling the IoT gaps there are not previous works in this direction, where a set of experiments have been conducted to test its reliability. In this manner, results of this paper not only show the convenience of this approach, but they also provide useful information about the most suitable CF techniques to be used in the IoT context, the relationship between sparsity level and accuracy, and the existing sparsity thresholds to switch from memory to model-based methods. We expect that the outcome of these experiments will be valuable for the scientific community for making IoT sensor arrays even more accurate and useful.

The conducted experiments clearly evidence that collaborative filtering techniques can be used to accurately predict failed sensor values from the whole set of sensor readings collected from the IoT network, confirming the paper’s hypothesis. Overall, accurate values have been obtained predicting missing sensor values and, as expected, accuracy increases as the sparsity level decreases. Looking more carefully at the obtained data, the best method for small sparsity rates (i.e., small failure rate) is RowKNN, while model-based methods like BiasedMF and NMF present a more robust and better performance for high sparsity.

This observation perfectly agrees with the underlying mathematical work mode of these methods. As its name suggests, RowKNN operates on rows, predicting the missing values from the *K* nearest neighbors. In this case, each row is the collection of measures of the whole sensor array at a certain instant of time, so RowKNN predicts missing values by looking at the most similar measurements at different times. In other words, RowKNN searches for array measurements at moments in which the environmental conditions of the measured medium were very similar to ones of the instant in which the missing reading happened (say, same phases of a chemical reaction, or same climatic conditions). The nature of the data, coming from sensor arrays that are measuring a close-to-deterministic physical phenomenon, leads to very accurate predictions since the system is just extrapolating data from similar states.

However, this approach is valid as long as the system is able to find another reading that took place under similar conditions than the one to be predicted. When the failure rate increases, the number of complete readings stored drastically decreases, so it harder for the system to identify similar environmental conditions and many phenomena remain undetected. There exists, thus, a threshold in the sparsity rate above which the richness and variety of the data is insufficient for the memory-based method, and the accuracy drastically decreases. Supporting this statement observe that, as shown in Figure 7, RowKNN reached a very good performance (outperforming the model-based methods) for much higher sparsity levels in the large sensor array dataset [40] than in the smaller sensor array dataset [39]. This is produced by the fact that, since [40] contains simultaneous readings from 128 sensors, even if there are some missing values in data array, the algorithm is able to identify the underlying conditions. On the contrary, when dealing with the dataset [39] with only 16 measures at once, a single missing value in the array makes it harder to compare with other readings, so the method is much sensitive to the sparsity level.

On the other hand, model-based CF methods show a much more robust and stable performance. As we explained in Section 2.1, these methods are specifically designed to extract hidden features of the phenomenon under consideration, giving rise an abstraction of the particular data. As we explained, in MF methods this abstract representation is encoded in the smaller dimensional matrix factorization. In this way, the memory-based methods are intrinsically ill when the number of available data decreases since they cannot find similar neighbors but, as long as the data is rich enough to encode these hidden features, the model-based methods are able to distillate them and to use them to issue much accurate predictions. For this reason, in Figure 7 we observe a better performance of the model-based methods with respect to the memory-based methods for high sparsity. It is also remarkable that all the model-based methods reached very similar performance values, straightening the soundness of the approach (with maybe the exception of the old-fashioned PMF method).

It is worthy to mention that, in sharp contrast with the good performance of RowKNN, the transpose method ColKNN outputs constantly very bad results in the predictions. The reason under these poor results lies in the structure of the datasets. Because of the way they were constructed, the datasets present many fewer columns (i.e., sensors in the array) than rows (i.e., measures). In the [40] dataset there are more than 10,000 readings against 128 columns, and the balance is even worse with the [39] dataset, where there are more than 80,000 rows against 16 columns. In this way, the number of neighbors under consideration is really small, so the method essentially degenerates to predict the missing values of a sensor with the readings of its “most similar” sensor at the same time. This is definitely a very naïve approach.

Another important contribution of this paper is the proposal of a method for generating datasets based on copulas. Due to the private scope of the IoT networks, and the exploitation rights of the providing companies, it is challenging to find large and structured datasets that are suitable to test machine learning methods. Trying to overcome this problem, in this paper we propose to import a statistical tool, the copulas, to infer the underlying joint probability distribution of a real dataset (maybe small and incomplete). This allows us to synthesize, de novo, a whole dataset of sensor measurements. Apart from obtaining a new testbench for our experiments, the synthetic dataset typically smooths out the outliers presented in the original dataset, providing a purer statistical representation of the data. As we show in Figure 8, the results of the proposed CF based RS methods on the synthetic dataset are perfectly compatible with the ones of the original dataset [39], and similar trends and performances are observed for the above-mentioned algorithms. Thereby, this experiment strengthens the reliability in our conclusions, since good performance of a method on both real and synthetic datasets evidences that the method is able to extract the underlying model and tents to be much more robust and to generalize to unexpected situations.

Regarding these results on the synthetic dataset, it a remarkable observation that, now, model-based methods outperform memory-based methods even for low sparsity scenarios. This is due to the fact that the synthetic dataset mimics the probability distribution of the original dataset, so model-based methods are still able to distill the data and to infer the model, giving rise to similar results on both datasets. However, when only the shallow probability distribution is captured, we are losing the deeper information about the underlying deterministic physical phenomena that are generating the environment measured by the sensors. In this way, even though the samples from the real and the synthetic dataset are very similar in a probabilistic sense, the synthetic samples lose some subtle information about the underlying phenomena that memory-based methods were able to extract from the real samples. This leads to an slightly worse performance of the memory-based methods in low sparsity scenarios on the synthetic dataset compared to the real one.

From the results shown in this paper, several lines of future work open to extend and refine the proposed methods. One of the most obvious one would be to test the proposed approach in other IoT networks and noisy scenarios, to evaluate the robustness of the method with different types of sensors and failure causes. Moreover, in this work we have focused on the CF methods at the boundary of the state-of-art in RSs, namely KNN-like methods and MF-based methods. Nevertheless, nowadays there exists an important trend in the recommender system field that aims to import techniques from deep learning to improve the CF methods (for example, by post-composing the MF with a deep neural network that breaks the linearity of the model). In this line, it would be interesting to test these avant-garde RS models in the IoT framework to elucidate whether this non-linearity could boost the results in high sparsity scenarios. Finally, another important research line to be explored the incorporation of the geometry of the sensor network to obtain finer topological information. This topology might be replicated in the underlying model to exploit the locality and neighbourship information of the IoT sensors geographic distribution, giving rise to an improvement of the predictions.

## Figures and Tables

**Figure 1 sensors-20-04628-f001:**
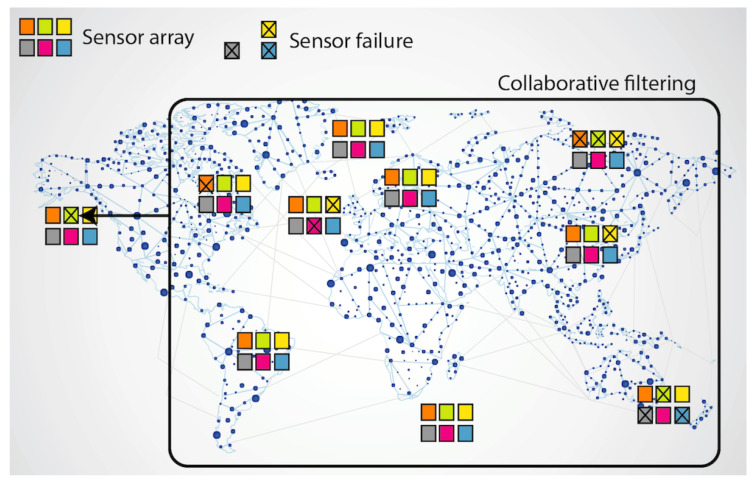
Collaborative filtering approach to predict sensors failed values.

**Figure 2 sensors-20-04628-f002:**
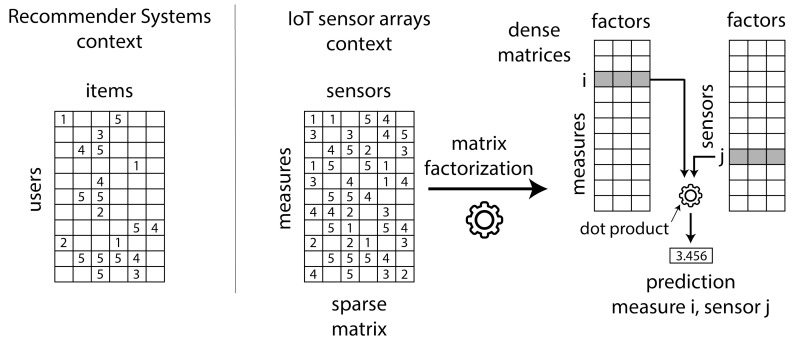
Collaborative filtering operation in the IoT sensor arrays context.

**Figure 3 sensors-20-04628-f003:**
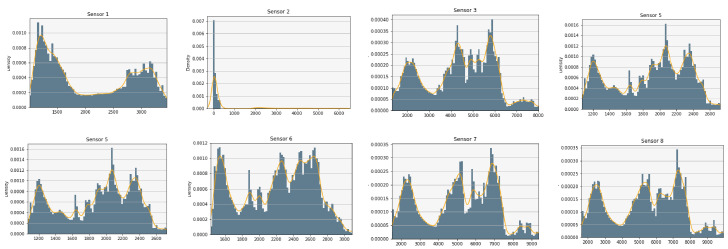
Distribution of the first 8 features in the [39] dataset. Each subplot shows the distribution of a reading of the sensor array. In grey, the histogram of the feature, and in orange a Gaussian kernel estimator of its density.

**Figure 4 sensors-20-04628-f004:**
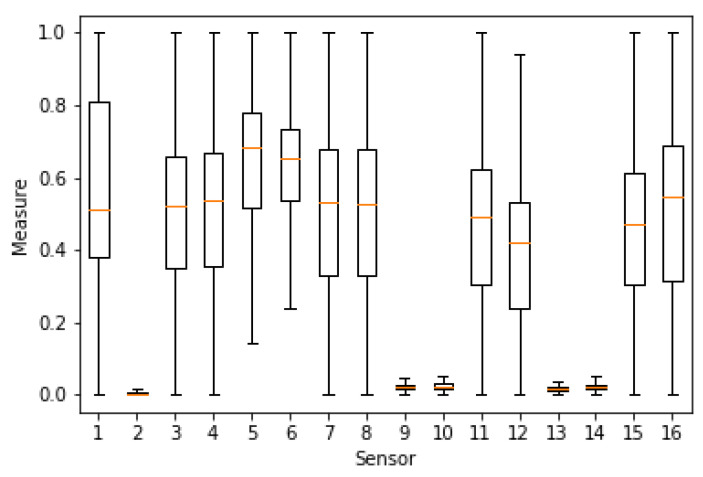
Boxplot of the normalized range of values for the 16 sensors in the [39] dataset. *x*-axis: sensor id, *y*-axis: range of results.

**Figure 5 sensors-20-04628-f005:**
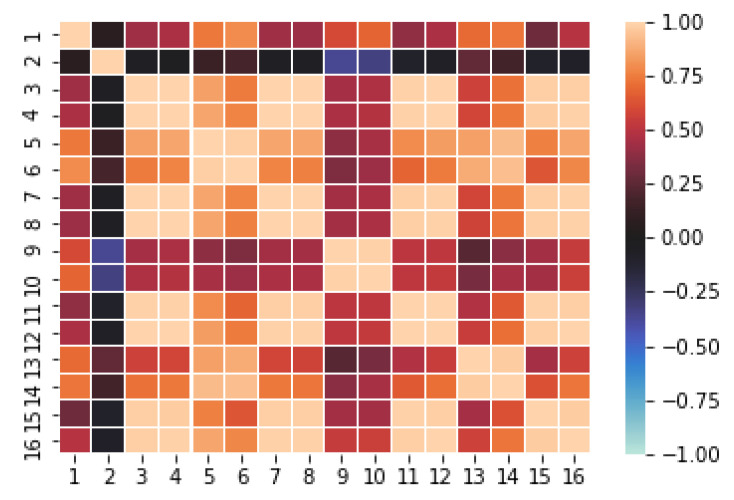
Correlation matrix for the 16 sensors in the [39] dataset. Warm colors (with maximum light orange) stand for strong positive correlation and cool colors (with maximum light blue) mean strong negative correlation. Black indicates no linear correlation.

**Figure 6 sensors-20-04628-f006:**
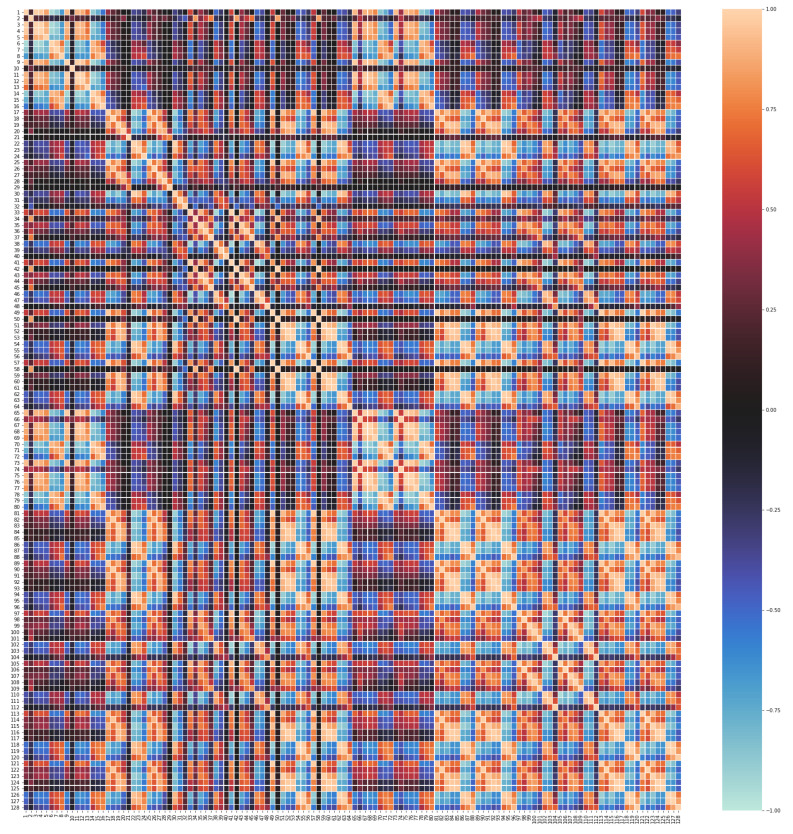
Correlation matrix for the 128 sensors in the [40] dataset. Warm colors (with maximum light orange) stand for strong positive correlation and cool colors (with maximum light blue) mean strong negative correlation. Black indicates no linear correlation.

**Figure 7 sensors-20-04628-f007:**
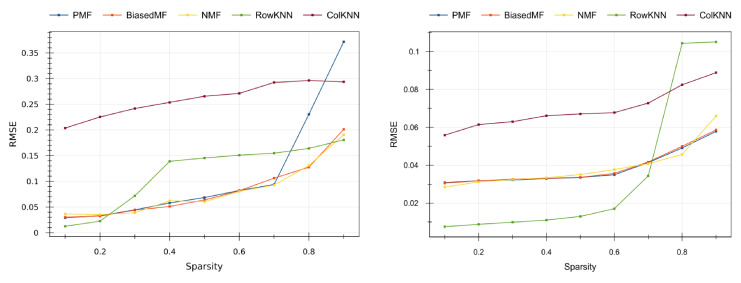
Prediction quality results. Left side: [39] dataset, right side: [40] dataset; *x*-axis: sparsity levels; *y*-axis: prediction error. Best results are the closest to zero (low errors).

**Figure 8 sensors-20-04628-f008:**
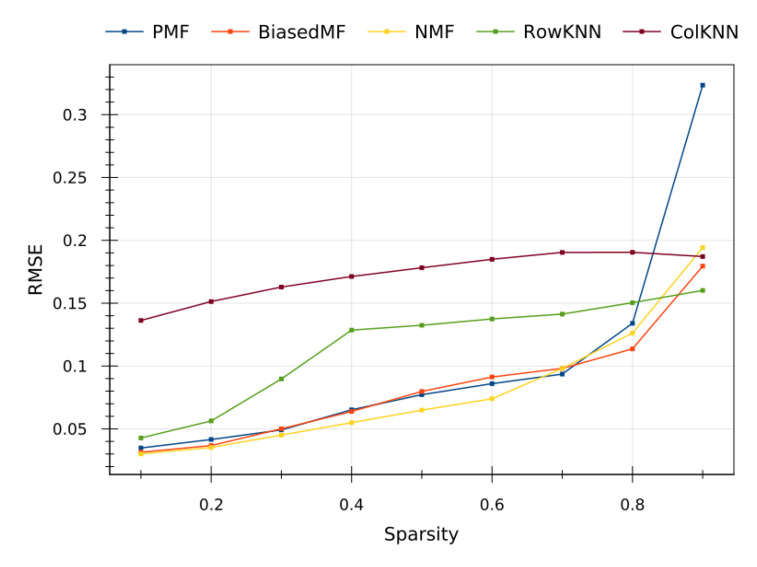
Prediction quality results obtained from the synthetic dataset.

**Table 1 sensors-20-04628-t001:** Most adequate hyper-parameters reported by a grid search optimization on the [39] dataset.

Sparsity	PMF			BiasedMF			NMF	RowKNN	ColKNN
	**Factors**	γ	λ	**Factors**	γ	λ	Factors	Neighbors	Neighbors
0.1	20	0.070	0.005	15	0.070	0.005	10	25	5
0.2	20	0.070	0.005	15	0.070	0.005	5	25	5
0.3	15	0.070	0.005	15	0.070	0.005	5	25	5
0.4	10	0.070	0.010	20	0.070	0.015	5	25	5
0.5	15	0.070	0.015	20	0.070	0.020	5	25	5
0.6	10	0.070	0.020	20	0.070	0.030	15	25	5
0.7	10	0.035	0.005	15	0.070	0.040	15	25	5
0.8	15	0.040	0.005	15	0.070	0.070	15	25	5
0.9	15	0.070	0.070	15	0.070	0.070	15	25	5

**Table 2 sensors-20-04628-t002:** Most adequate hyper-parameters reported by a grid search optimization on the [40] dataset.

Sparsity	PMF			BiasedMF			NMF	RowKNN	ColKNN
	**Factors**	γ	λ	**Factors**	γ	λ	Factors	Neighbors	Neighbors
0.1	20	0.045	0.005	20	0.045	0.005	20	25	5
0.2	20	0.050	0.005	20	0.050	0.005	20	25	5
0.3	20	0.055	0.005	20	0.060	0.005	20	25	5
0.4	20	0.060	0.005	20	0.065	0.005	15	25	5
0.5	20	0.070	0.005	20	0.070	0.005	20	25	5
0.6	20	0.070	0.005	15	0.070	0.005	20	25	5
0.7	20	0.070	0.005	20	0.070	0.005	5	25	5
0.8	5	0.070	0.005	20	0.070	0.010	5	25	5
0.9	5	0.070	0.005	5	0.070	0.010	5	25	5

**Table 3 sensors-20-04628-t003:** Most adequate hyper-parameters reported by a grid search optimization on the synthetic dataset.

Sparsity	PMF			BiasedMF			NMF	RowKNN	ColKNN
	**Factors**	γ	λ	**Factors**	γ	λ	Factors	Neighbors	Neighbors
0.1	20	0.070	0.005	20	0.070	0.005	5	25	5
0.2	20	0.070	0.005	20	0.070	0.005	5	25	5
0.3	20	0.070	0.005	20	0.070	0.010	10	25	5
0.4	15	0.070	0.015	20	0.070	0.015	10	25	5
0.5	15	0.070	0.020	10	0.070	0.020	5	25	5
0.6	10	0.030	0.005	10	0.070	0.030	15	25	5
0.7	10	0.040	0.005	10	0.002	0.065	10	25	5
0.8	10	0.045	0.005	10	0.030	0.070	15	25	5
0.9	10	0.070	0.070	15	0.070	0.070	20	25	5

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
