# Peer review of "Collaborative Filtering to Predict Sensor Array Values in Large IoT Networks"

_sensors, 2020, doi:10.3390/s20164628_

Round 1
Reviewer 1 Report
1. "Since the sparsity of the IoT failures is not as higher as the CF datasets, we have selected both memory-based and model-based methods as baselines to make the experiments. From the memory-based algorithms we have chosen the representative KNN method based on items [21] and the KNN based on users [24]. From the model-based methods we have chosen the PMF [29], BiasedMF [30] and NMF [31] matrix factorization ones." - It is not explained why these methods are selected. What criteria is used for such selection?
2. Tables 1-3 are experimental settings, not results.
3. Figure 8 should be placed in the results section, not in conclusions. It is mentioned in the Results section.
4. References could be improved (some are missing for Introduction section)
5. "In this paper, we have shown that collaborative filtering approaches can be used to predict missing values from sensor arrays deployed in large IoT systems, produced due to n reliable sensors, extreme environmental situations, deficient QoS and many other problematic issues. Large IoT networks provide to the cloud the necessary information to fill the sensor reading gaps from the existing sensor values. This is possible due to the underlying correlations between sensor arrays deployed in large numbers of different geographical spots." - It looks like discovering hot water. It is reasonable to expect this. It should be presented why that result is important for scientific community.
Author Response
Please see the attachment.
Yours faithfully,
F. Ortega, A. González-Prieto, J. Bobadilla and A. Gutiérrez

Reviewer 2 Report
This paper proposes to use collaborative filtering techniques to predict missing readings. I think this paper needs extensive work before it can be published in the sensors journal, so I recommend a clear rejection.
- The experimental results section must be extended and each experiment must be explained more carefully.
- The dataset description was mixed in the methodology. It should be put in one section or subsection.
- Please divide section 2 into smaller subsections, each subsection describes one specific topic. The current manuscript is very hard to follow
- Regarding the equations, many variables are not defined. Please define all variables.
- The comparison with recent techniques can better show the advantage of the proposed method.
- The discussion and conclusion appear to be just a summary of work done, rather than a critical appraisal of the presented research work. When the proposed algorithm outperforms others, why is this the case? Any insights what makes this algorithm accurate, from a mathematical/algorithmic point of view? Any weaknesses?
Author Response

(The authors gave the same response as above.)

Round 2
Reviewer 1 Report
Correlation and correlation matrix could be defined and expressions added.
Describe role of colors in the correlation matrix (figs 5,6). What does mean black, red, etc.? Comment groups of neighbors of the same color.
Author Response
Thank you very much for your comment. We've added some extra information in the caption of Figures 5 and 6, explaining the colour code of the heat map. We also included a paragraph with a discussion about some of the correlation clusters found in both dataset and the existence of isolated sensors. The paragraph reads as follows:
"For instance, in [39] (Figure 5), we find that sensors 3, 4, 7, 8, 11, 12, 15, 16 form a highly intra-correlated cluster, with strong positive correlations. There exists another cluster formed by the sensors 1, 5, 6, 9, 10, 13, 14 with smaller positive correlation inter and intra-cluster. On the other hand, sensor 2 presents a much weaker correlation with the other sensors (except maybe with 9 and 10) which suggests that sensor 2 has a different nature than the other ones in the dataset. Similar patterns can be observed in [40] (Figure 6), where, roughly speaking, sensors 1-16 and 65-80 seem to form a highly intra-correlated cluster, with lower inter-correlation. Observe that, due to the organization of the dataset, readings 1-16 correspond to the 8 measured features of the first 2 sensors, while readings 65-80 correspond to the 8 features of sensors 9 and 10. In addition, as in [39], we also observe isolated sensors, like sensors 10, 21 or 29, with very low absolute correlation with any other sensor."
Following your suggestion, we also included a brief review of the description of the correlation between two random variables.
Please, if you consider that further modifications are needed, just let us know.
Reviewer 2 Report
All of my comments have been addressed. Thus, I recommend this paper for publication
Author Response
Thank you very much for your words and careful reading of the manuscript. We are glad you liked the new version.